# 3D Diffuser Actor: Policy Diffusion with 3D Scene Representations

**Tsung-Wei Ke**[†]      **Nikolaos Gkanatsios**[†]      **Katerina Fragkiadaki**

Carnegie Mellon University

{tsungwek,ngkanats,katef}@cs.cmu.edu

3d-diffuser-actor.github.io

**Abstract:** Diffusion policies are conditional diffusion models that learn robot action distributions conditioned on the robot and environment state. They have recently shown to outperform both deterministic and alternative action distribution learning formulations. 3D robot policies use 3D scene feature representations aggregated from a single or multiple camera views using sensed depth. They have shown to generalize better than their 2D counterparts across camera viewpoints. We unify these two lines of work and present 3D Diffuser Actor, a neural policy equipped with a novel 3D denoising transformer that fuses information from the 3D visual scene, a language instruction and proprioception to predict the noise in noised 3D robot pose trajectories. 3D Diffuser Actor sets a new state-of-the-art on RLBench with an absolute performance gain of 18.1% over the current SOTA on a multi-view setup and an absolute gain of 13.1% on a single-view setup. On the CALVIN benchmark, it improves over the current SOTA by a 9% relative increase. It also learns to control a robot manipulator in the real world from a handful of demonstrations. Through thorough comparisons with the current SOTA policies and ablations of our model, we show 3D Diffuser Actor's design choices dramatically outperform 2D representations, regression and classification objectives, absolute attentions, and holistic non-tokenized 3D scene embeddings.

## 1 Introduction

Many robot manipulation tasks are inherently multimodal: at any point during task execution, there may be multiple actions which yield task-optimal behavior. Indeed, human demonstrations often contain diverse ways that a task can be accomplished. A natural choice is then to treat policy learning as a distribution learning problem: instead of representing a policy as a deterministic map $\pi_\theta(x)$, learn the entire distribution of actions conditioned on the current robot state $p(y|x)$ [1, 2, 3, 4].

Recent works use diffusion objectives for learning such state-conditioned action distributions for robot manipulation policies from demonstrations [5, 6, 7]. They outperform deterministic or other alternatives, such as variational autoencoders [8], mixture of Gaussians [9], combination of classification and regression objectives [4], or energy-based objectives [10]. They typically use either low-dimensional (oracle) states [5] or 2D images [6] as their scene representation.

At the same time, 3D robot policies build scene representations by "lifting" features from perspective views to a 3D robot workspace based on sensed depth and camera extrinsics [11, 12, 13, 14, 15, 16]. They have shown to generalize better than 2D robot policies across camera viewpoints and to handle novel camera viewpoints at test time [15, 16]. We conjecture this improved performance comes from the fact that the visual scene tokens and the robot's actions interact in a common 3D space, that is robust to camera viewpoints, while in 2D policies the neural network need to learn the 2D-to-3D mapping implicitly.

In this work, we marry diffusion for handling multimodality in action prediction with 3D scene representations for effective spatial reasoning. We propose 3D Diffuser Actor, a novel 3D denoising

8th Conference on Robot Learning (CoRL 2024), Munich, Germany.

policy transformer that takes as input a tokenized 3D scene representation, a language instruction and a noised end-effector's future translation and rotation trajectory, and predicts the error in translations and rotations for the robot's end-effector. The model represents both the scene tokens and the end-effector locations in the same 3D space and fuses them with relative-position 3D attentions [17, 18], which achieves translation equivariance and helps generalization.

We test 3D Diffuser Actor in learning robot manipulation policies from demonstrations on the simulation benchmarks of RLBench [19] and CALVIN [20], as well as in the real world. 3D Diffuser Actor sets a new state-of-the-art on RLBench with a 18.1% absolute gain on multi-view setups and 13.1% on single-view setups, outperforming existing 3D policies and 2D diffusion policies. On CALVIN, it outperforms the current SOTA in the setting of zero-shot unseen scene generalization by a 9% relative gain. We further show 3D Diffuser Actor can learn multi-task manipulation in the real world across 12 tasks from a handful of real-world demonstrations. We empirically show that 3D Diffuser Actor outperforms all existing policy formulations, which either do not use 3D scene representations, or do not use action diffusion. We further compare against ablative versions of our model and show the importance of the 3D relative attentions.

**Our contributions:** The main contribution of this work is to combine 3D scene representations and diffusion objectives for learning robot policies from demonstrations. 3D robot policies have not yet been combined with diffusion objectives. An exception is ChainedDiffuser [21], that uses diffusion models as a drop-in replacement for motion planners, rather than manipulation policies, since it relies on other learning-based policies (Act3D [15]) to supply the target 3D keypose to reach. We compare against ChainedDiffuser in our experiments and show we greatly outperform it.

**Concurrent work:** Concurrent to our effort, 3D diffusion policy [22] shares a similar goal of combining 3D representations with diffusion objectives for learning manipulation from demonstrations. Though the two works share the same goal, they have very different architectures. Unlike 3D Diffuser Actor, the model of [22] does not condition on a tokenized 3D scene representation but rather on a holistic 1D embedding pooled from the 3D scene point cloud. We compare 3D Diffuser Actor against [22] in our experiments and show it greatly outperforms it. We believe this is because tokenized scene representations, used in 3D Diffuser Actor, are robust to scene changes: if a part of the scene changes, only the corresponding subset of 3D scene tokens is affected. In contrast, holistic scene embeddings pooled across the scene are always affected by any scene change. Thanks to this spatial disentanglement of the 3D scene tokenization, 3D Diffuser Actor generalizes better.

Code and videos of our manipulation results are available at `https://3d-diffuser-actor.github.io/`.

## 2  Related Work

**Learning manipulation policies from demonstrations** Earlier works on learning from demonstrations train deterministic policies with behavior cloning [23, 24]. To better handle action multimodality, approaches discretize action dimensions and use cross entropy losses [25, 14, 11]. Generative adversarial networks [1, 2, 26], variational autoencoders [8], combined Categorical and Gaussian distributions [4, 27, 28] and Energy-Based Models (EBMs) [10, 29, 30] have been used to learn from multimodal demonstrations. Diffusion models [31, 32] are a powerful class of generative models related to EBMs in that they model the score of the distribution, else, the gradient of the energy, as opposed to the energy itself [33, 34]. The key idea behind diffusion models is to iteratively transform a simple prior distribution into a target distribution by applying a sequential denoising process. They have been used for modeling state-conditioned action distributions in imitation learning [35, 36, 5, 37, 7, 38] from low-dimensional input, as well as from visual sensory input, and show both better mode coverage and higher fidelity in action prediction than alternatives.

**Diffusion models in robotics** Beyond policy representations in imitation learning, diffusion models have been used to model cross-object and object-part arrangements [39, 40, 38, 41, 30], visual image subgoals [42, 43, 44, 45], and offline reinforcement learning [46, 47, 48]. ChainedDiffuser [21]

proposes to replace motion planners, commonly used for keypose to keypose linking, with a trajectory diffusion model that conditions on the 3D scene feature cloud and the predicted target 3D keypose to denoise a trajectory from the current to the target keypose. It uses a diffusion model that takes as input 3D end-effector keyposes predicted by Act3D [15] and a 3D representation of the scene to infer robot end-effector trajectories that link the current end-effector pose to the predicted one. 3D Diffuser Actor instead predicts the next 3D keypose for the robot's end-effector alongside the linking trajectory, which is a much harder task than linking two given keyposes. 3D Diffusion Policy [22] also combines 3D scene representations with diffusion objectives but uses 1D point cloud embeddings. We compare against both ChainedDiffuser and 3D Diffusion Policy in our experimental section and show we greatly outperform them.

**2D and 3D scene representations for robot manipulation** End-to-end image-to-action policy models, such as RT-1 [49], RT-2 [50], GATO [51], BC-Z [52], RT-X [53], Octo [54] and InstructRL [55] leverage transformer architectures for the direct prediction of 6-DoF end-effector poses from 2D video input. However, this approach comes at the cost of requiring thousands of demonstrations to implicitly model 3D geometry and adapt to variations in the training domains. 3D scene-to-action policies, exemplified by C2F-ARM [13] and PerAct [14], involve voxelizing the robot's workspace and learning to identify the 3D voxel containing the next end-effector keypose. However, this becomes computationally expensive as resolution requirements increase. Consequently, related approaches resort to either coarse-to-fine voxelization, equivariant networks [12] or efficient attention operations [56] to mitigate computational costs. Act3D [15] foregoes 3D scene voxelization altogether; it instead computes a 3D action map of variable spatial resolution by sampling 3D points in the empty workspace and featurizing them using cross-attentions to the 3D physical scene points. Robotic View Transformer (RVT) [16] re-projects the input RGB-D image to alternative image views, featurizes those and lifts the predictions to 3D to infer 3D locations for the robot's end-effector.

3D Diffuser Actor builds upon works of Act3D [15] from 3D policies and upon the works of [36, 6] from diffusion policies. It uses a tokenized 3D scene representation, similar to [15], but it is a probabilistic model instead of a deterministic one. It does not sample 3D points and does not infer 3D action maps. It uses diffusion objectives instead of classification or regression objectives used in [15]. In contrast to [40, 36], it uses 3D scene representations instead of 2D images or low-dimensional states. We compare against both 2D diffusion policies and 3D policies in our experiments and show 3D Diffuser Actor greatly outperforms them. We highlight the differences between our model and related models in Figure 3 in the Appendix, and we refer to Figures 4 and 5 for more architectural details of 3D Diffuser Actor and Act3D.

## 3   Method

3D Diffuser Actor is trained to imitate demonstration trajectories of the form of $\{(\mathbf{o}_1, \mathbf{a}_1), (\mathbf{o}_2, \mathbf{a}_2), ...\}$, accompanied with a task language instruction $l$, similar to previous works [57, 14, 15, 58], where $\mathbf{o}_t$ stands for the visual observation and $\mathbf{a}_t$ stands for robot action at timestep $t$. Each observation $\mathbf{o}_t$ is one or more posed RGB-D images. Each action $\mathbf{a}_t$ is an end-effector pose and is decomposed into 3D location, rotation and binary (open/close) state: $\mathbf{a}_t = \{\mathbf{a}_t^{\mathrm{loc}} \in \mathbb{R}^3, \mathbf{a}_t^{\mathrm{rot}} \in \mathbb{R}^6, \mathbf{a}_t^{\mathrm{open}} \in \{0,1\}\}$. We represent rotations using the 6D rotation representation of [59] for all environments in all our experiments, to avoid the discontinuities of the quaternion representation. We will use the notation $\boldsymbol{\tau}_t = (\mathbf{a}_{t:t+T}^{\mathrm{loc}}, \mathbf{a}_{t:t+T}^{\mathrm{rot}})$ to denote the trajectory of 3D locations and rotations at timestep $t$, of temporal horizon $T$. Our model, at each timestep $t$ predicts a trajectory $\boldsymbol{\tau}_t$ and binary states $\mathbf{a}_{t:t+T}^{\mathrm{open}}$.

The architecture of 3D Diffuser Actor is shown in Figure 1. It is a conditional diffusion probabilistic model [31, 60] of trajectories given the visual scene and a language instruction; it predicts a whole trajectory $\boldsymbol{\tau}$ at once, non autoregressively, through iterative denoising, by inverting a process that gradually adds noise to a sample $\boldsymbol{\tau}^0$. The diffusion process is associated with a variance schedule $\{\beta^i \in (0,1)\}_{i=1}^N$, which defines how much noise is added at each diffusion step. The noisy version of sample $\boldsymbol{\tau}^0$ at diffusion step $i$ can then be written as $\boldsymbol{\tau}^i = \sqrt{\bar{\alpha}^i}\boldsymbol{\tau}^0 + \sqrt{1 - \bar{\alpha}^i}\epsilon$, where $\epsilon \sim \mathcal{N}(\mathbf{0}, \mathbf{1})$

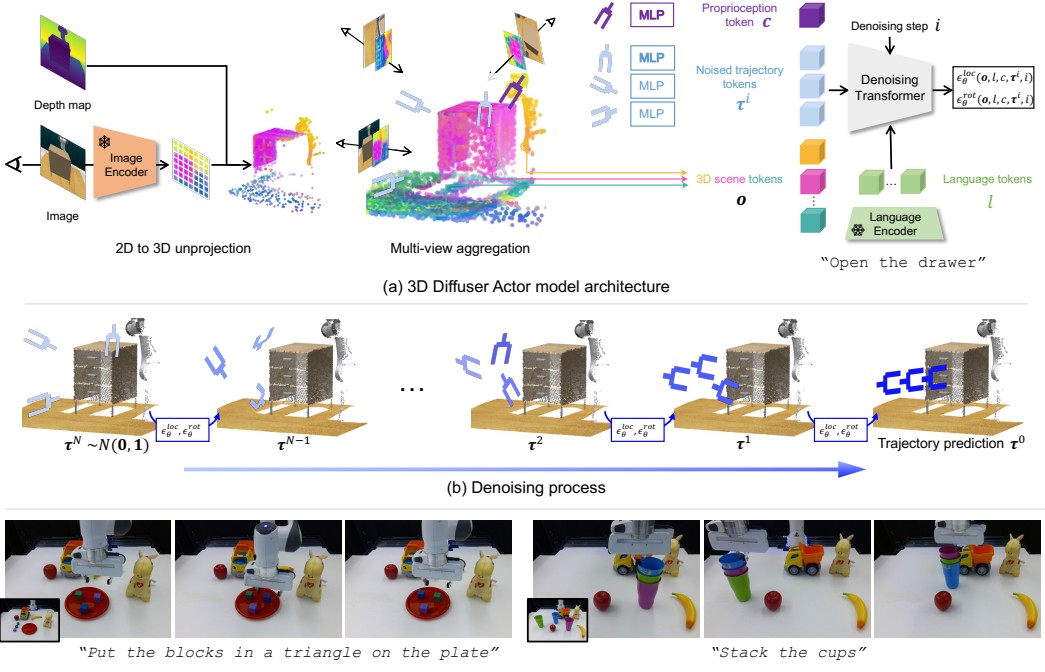

(a) 3D Diffuser Actor model architecture

(b) Denoising process

(c) Real-world multi-task manipulation tasks

Figure 1: **Overview of 3D Diffuser Actor**. **(a)** 3D Diffuser Actor is a conditional diffusion probabilistic model of robot 3D trajectories. At diffusion step $i$, it converts the current noised estimate of the robot's future action trajectory $\boldsymbol{\tau}^i$, the posed RGB-D views $\mathbf{o}$, and proprioceptive information $c$ to a set of 3D tokens. It fuses information among these 3D tokens and language instruction tokens $l$ using 3D relative denoising transformers to predict the noise of 3D robot locations $\epsilon_\theta^{loc}(\mathbf{o}, l, c, \boldsymbol{\tau}^i, i)$ and the noise of 3D robot rotations $\epsilon_\theta^{rot}(\mathbf{o}, l, c, \boldsymbol{\tau}^i, i)$. **(b)** During inference, 3D Diffuser Actor iteratively denoises the noised estimate of the robot's future trajectory. **(c)** 3D Diffuser Actor works in the real world and captures the multiple behavioural modes in the training demonstrations.

is a sample from a Gaussian distribution (with the same dimensionality as $\boldsymbol{\tau}^0$), $\alpha^i = 1 - \beta^i$, and $\bar{\alpha}^i = \prod_{j=1}^i \alpha^j$.

**3D Relative Denoising Transformer** 3D Diffuser Actor models a learned gradient of the denoising process with a 3D relative transformer $\hat{\epsilon} = \epsilon_\theta(\boldsymbol{\tau}_t^i; i, \mathbf{o}_t, l, c_t)$ that takes as input the noisy trajectory $\boldsymbol{\tau}_t^i$ at timestep $t$, diffusion step $i$, and conditioning information from the language instruction $l$, the visual observation $\mathbf{o}_t$ and proprioception $c_t$ of timestep $t$, to predict the noise component $\hat{\epsilon}$. At each timestep $t$ and diffusion step $i$, we convert visual observations $\mathbf{o}_t$, proprioception $c_t$ and noised trajectory estimate $\boldsymbol{\tau}_t^i$ to a set of 3D tokens. Each 3D token is represented by a latent embedding and a 3D position. Our model fuses all 3D tokens using relative 3D attentions, and additionally fuses information from the language instruction using normal attentions, since it is meaningless to define 3D coordinates for language tokens. Next, we describe how we featurize each piece of input (when not ambiguous, we omit the subscript $t$ for clarity).

**3D tokenization** At each diffusion step $i$, we represent the noisy estimate $\boldsymbol{\tau}^i$ of the clean trajectory $\boldsymbol{\tau}^0$ as a sequence of 3D trajectory tokens, by mapping each noisy pose $\mathbf{a}^i$ of $\boldsymbol{\tau}^i$, to a latent embedding vector with a MLP, and a 3D position which comes from the noised 3D translation component $\mathbf{a}^{loc,i}$. We featurize each image view using a 2D feature extractor and obtain a corresponding 2D feature map $F \in \mathbb{R}^{H \times W \times c}$, where $c$ is the number of feature channels and $H, W$ the spatial dimensions, using a pre-trained CLIP ResNet50 2D image encoder [61]. Given the corresponding depth map from that view, we compute the 3D location $(X, Y, Z)$ for each of the $H \times W$ feature patches by averaging the depth value inside the patch extent. We map the pixel coordinate and depth value of a patch to a corresponding 3D coordinate using camera intrinsics and extrinsics and the pinhole camera model. This results in a 3D scene token set of cardinality $H \times W$. Each scene token is

represented by the corresponding patch feature vector $F_{x,y}$, the feature vector at the corresponding patch coordinate $(x, y)$, and a 3D position. If more than one views are available, we aggregate 3D scene tokens from each, to obtain the final 3D scene token set. The proprioception $c$ is also a 3D scene token, with a learnable latent representation and the 3D positional embedding corresponding to the end-effector's current 3D location. Lastly, we map the language task instruction to language tokens using a pre-trained CLIP language encoder, following previous works [15].

Our 3D Relative Denoising Transformer applies relative self-attentions among all 3D tokens, and cross-attentions to the language tokens. For the 3D self-attentions, we use the rotary positional embeddings [18] to encode relative positional information in attention layers. The attention weight between query $q$ and key $k$ is written as: $e_{q,k} \propto \mathbf{x}_q^T \mathbf{M}(\mathbf{p}_q - \mathbf{p}_k)\mathbf{x}_k$, where $\mathbf{x}_q$ / $\mathbf{x}_k$ and $\mathbf{p}_q$ / $\mathbf{p}_k$ denote the features and 3D positions of the query / key, and $\mathbf{M}$ is a matrix function which depends only on the relative positions of the query and key, inspired by recent work in visual correspondence [62, 63] and 3D manipulation [15, 21]. We feed the final trajectory tokens to MLPs to predict: (1) the noise $\epsilon_\theta^{loc}(\mathbf{o}, l, c, \boldsymbol{\tau}^i, i)$ and $\epsilon_\theta^{rot}(\mathbf{o}, l, c, \boldsymbol{\tau}^i, i)$ added to $\boldsymbol{\tau}^0$'s sequence of 3D translations and 3D rotations, respectively, and (2) the end-effector opening $f_\theta^{\text{open}}(\mathbf{o}, l, c, \boldsymbol{\tau}^i, i) \in [0, 1]^T$.

**Training and inference** During training, we randomly sample a time step $t$ and a diffusion step $i$ and add noise $\epsilon = (\epsilon^{\text{loc}}, \epsilon^{\text{rot}})$ to a ground-truth trajectory $\boldsymbol{\tau}_t^0$. We use the $L1$ loss for reconstructing the sequence of 3D locations and 3D rotations. We use binary cross-entropy (BCE) loss to supervise the end-effector opening $f_\theta^{\text{open}}$, we use the prediction from i=1 at inference time. Our objective reads:

$$\mathcal{L}_\theta = w_1 \|(\epsilon_\theta^{\text{loc}}(\mathbf{o}, l, c, \boldsymbol{\tau}^i, i) - \epsilon^{\text{loc}}\| + w_2 \|(\epsilon_\theta^{\text{rot}}(\mathbf{o}, l, c, \boldsymbol{\tau}^i, i) - \epsilon^{\text{rot}}\| + \text{BCE}(f_\theta^{\text{open}}(\mathbf{o}, l, c, \boldsymbol{\tau}^i, i), \mathbf{a}_{1:T}^{\text{open}}),$$

where $w_1, w_2$ are hyperparameters estimated using cross-validation. To draw a sample from the learned distribution $p_\theta(\boldsymbol{\tau}|\mathbf{o}, l, c)$, we start by drawing a sample $\boldsymbol{\tau}_N \sim \mathcal{N}(\mathbf{0}, \mathbf{1})$. Then, we progressively denoise the sample by iterated application of $\epsilon_\theta$ $N$ times according to a specified sampling schedule [32, 64], which terminates with $\boldsymbol{\tau}_0$ sampled from $p_\theta(\boldsymbol{\tau})$: $\boldsymbol{\tau}^{i-1} = \frac{1}{\sqrt{\alpha^i}} \left( \boldsymbol{\tau}^i - \frac{\beta^i}{\sqrt{1-\bar{\alpha}_i}} \epsilon_\theta(\mathbf{o}, l, c, \boldsymbol{\tau}^i, i) \right) + \frac{1-\bar{\alpha}^{i+1}}{1-\bar{\alpha}^i} \beta^i \mathbf{z}$, where $\mathbf{z} \sim \mathcal{N}(\mathbf{0}, \mathbf{1})$ is a random variable of appropriate dimension. Empirically, we found that using separate noise schedulers for $\mathbf{a}^{\text{loc}}$ and $\mathbf{a}^{\text{rot}}$, specifically, scaled-linear and square cosine scheduler, respectively, achieves better performance.

**Implementation details** At training, we segment demonstration trajectories at detected end-effector *keyposes*, such as a change in the open/close end-effector state or local extrema of velocity/acceleration, following previous works [57, 14, 15, 55]. We then resample each trajectory segment to have the same length $T$. During inference, 3D Diffuser Actor can either predict and execute the full trajectory of actions up to the next keypose (including the keypose), or just predict the next keypose and use a sampling-based motion planner to reach it, similar to previous works [14, 27, 15].

Due to space limits, please check Section B.3 for detailed model diagram, Section B.5 for our choice of hyper-parameters, Section B.6 for detailed formulation of denoising diffusion probabilistic models and Section B.7 for discussion of noise schedulers in the Appendix.

## 4 Experiments

We test 3D Diffuser Actor in multi-task manipulation on RLBench [19] and CALVIN [20], two established learning from demonstrations benchmarks, and in the real world.

### 4.1 Evaluation on RLBench

RLBench is built atop the CoppelaSim [65] simulator, where a Franka Panda Robot is used to manipulate the scene. On RLBench, our model and all baselines are trained to predict the next end-effector keypose as opposed to keypose trajectory; all methods employ a low-level motion planner BiRRT [66], native to RLBench, to reach the predicted robot keypose. We train and evaluate 3D Diffuser Actor on two setups based on the number of available cameras: **1.** *Multi-view setup*, introduced in [14], that uses a suite of 18 manipulation tasks, each with 2-60 variations, which concern scene variability across object poses, appearance and semantics. There are four RGB-D cameras available, front, wrist, left shoulder and right shoulder. The wrist camera moves during manipulation. **2.** *Single-view setup*, introduced in [67], that uses a suite of 10 manipulation tasks.

| | Avg. Success ↑ | open drawer | slide block | sweep to dustpan | meat off grill | turn tap | put in drawer | close jar | drag stick | stack blocks |
|---|---|---|---|---|---|---|---|---|---|---|
| C2F-ARM-BC [13] | 20.1 | 20 | 16 | 0 | 20 | 68 | 4 | 24 | 24 | 0 |
| PerAct [14] | 49.4 | $88.0_{\pm5.7}$ | $74.0_{\pm13.0}$ | $52.0_{\pm0.0}$ | $70.4_{\pm2.0}$ | $88.0_{\pm4.4}$ | $51.2_{\pm4.7}$ | $55.2_{\pm4.7}$ | $89.6_{\pm4.1}$ | $26.4_{\pm3.2}$ |
| HiveFormer [27] | 45 | 52 | 64 | 28 | **100** | 80 | 68 | 52 | 76 | 8 |
| PolarNet [68] | 46 | 84 | 56 | 52 | **100** | 80 | 32 | 36 | 92 | 4 |
| RVT [16] | 62.9 | $71.2_{\pm6.9}$ | $81.6_{\pm5.4}$ | $72.0_{\pm0.0}$ | $88.0_{\pm2.5}$ | $93.6_{\pm4.1}$ | $88.0_{\pm5.7}$ | $52.0_{\pm2.5}$ | $99.2_{\pm1.6}$ | $28.8_{\pm3.9}$ |
| Act3D [15] | 63.2 | $78.4_{\pm11.2}$ | $96.0_{\pm2.5}$ | $\mathbf{86.4}_{\pm6.5}$ | $95.2_{\pm1.6}$ | $94.4_{\pm2.0}$ | $91.2_{\pm6.9}$ | $\mathbf{96.8}_{\pm3.0}$ | $80.8_{\pm6.4}$ | $4.0_{\pm3.6}$ |
| 3D Diffuser Actor (ours) | **81.3** | $\mathbf{89.6}_{\pm4.1}$ | $\mathbf{97.6}_{\pm3.2}$ | $84.0_{\pm4.4}$ | $\mathbf{96.8}_{\pm1.6}$ | $\mathbf{99.2}_{\pm1.6}$ | $\mathbf{96.0}_{\pm3.6}$ | $96.0_{\pm2.5}$ | $\mathbf{100.0}_{\pm0.0}$ | $\mathbf{68.3}_{\pm3.3}$ |

| | screw bulb | put in safe | place wine | put in cupboard | sort shape | push buttons | insert peg | stack cups | place cups | |
|---|---|---|---|---|---|---|---|---|---|---|
| C2F-ARM-BC [13] | 8 | 12 | 8 | 0 | 8 | 72 | 4 | 0 | 0 | |
| PerAct [14] | $17.6_{\pm2.0}$ | $86.0_{\pm3.6}$ | $44.8_{\pm7.8}$ | $28.0_{\pm4.4}$ | $16.8_{\pm4.7}$ | $92.8_{\pm3.0}$ | $5.6_{\pm4.1}$ | $2.4_{\pm2.2}$ | $2.4_{\pm3.2}$ | |
| HiveFormer [27] | 8 | 76 | 80 | 32 | 8 | 84 | 0 | 0 | 0 | |
| PolarNet [68] | 44 | 84 | 40 | 12 | 12 | 96 | 4 | 8 | 0 | |
| RVT [16] | $48.0_{\pm5.7}$ | $91.2_{\pm3.0}$ | $91.0_{\pm5.2}$ | $49.6_{\pm3.2}$ | $36.0_{\pm2.5}$ | $\mathbf{100.0}_{\pm0.0}$ | $11.2_{\pm3.0}$ | $26.4_{\pm8.2}$ | $4.0_{\pm2.5}$ | |
| Act3D [15] | $32.8_{\pm6.9}$ | $95.2_{\pm4.0}$ | $59.2_{\pm9.3}$ | $67.2_{\pm3.0}$ | $29.6_{\pm3.2}$ | $93.6_{\pm2.0}$ | $24.0_{\pm8.4}$ | $9.6_{\pm6.0}$ | $3.2_{\pm3.0}$ | |
| 3D Diffuser Actor (ours) | $\mathbf{82.4}_{\pm2.0}$ | $\mathbf{97.6}_{\pm2.0}$ | $\mathbf{93.6}_{\pm4.8}$ | $\mathbf{85.6}_{\pm4.1}$ | $\mathbf{44.0}_{\pm4.4}$ | $98.4_{\pm2.0}$ | $\mathbf{65.6}_{\pm4.1}$ | $\mathbf{47.2}_{\pm8.5}$ | $\mathbf{24.0}_{\pm7.6}$ | |

Table 1: **Evaluation on RLBench on the multi-view setup.** We show the mean and standard deviation of success rates average across all random seeds. 3D Diffuser Actor outperforms all prior arts on most tasks by a large margin. Variances are included when available.

| | Avg. Success ↑ | close jar | open drawer | sweep to dustpan | meat off grill | turn tap | slide block | put in drawer | drag stick | push buttons | stack blocks |
|---|---|---|---|---|---|---|---|---|---|---|---|
| GNFactor [67] | 31.7 | 25.3 | 76.0 | 28.0 | 57.3 | 50.7 | 20.0 | 0.0 | 37.3 | 18.7 | 4.0 |
| Act3D [15] | 65.3 | $52.0_{\pm5.7}$ | $84.0_{\pm8.6}$ | $80.0_{\pm9.8}$ | $66.7_{\pm1.9}$ | $64.0_{\pm5.7}$ | $\mathbf{100.0}_{\pm0.0}$ | $54.7_{\pm3.8}$ | $86.7_{\pm1.9}$ | $64.0_{\pm1.9}$ | $0.0_{\pm0.0}$ |
| 2D Diffuser Actor | 8.4 | $0_{\pm0}$ | $0_{\pm0}$ | $0_{\pm0}$ | $4.0_{\pm3.3}$ | $56.0_{\pm3.3}$ | $16.0_{\pm6.5}$ | $0_{\pm0}$ | $0_{\pm0}$ | $8.0_{\pm3.3}$ | $0_{\pm0}$ |
| 3D Diffuser Actor | 78.4 | $\mathbf{82.7}_{\pm1.9}$ | $\mathbf{89.3}_{\pm7.5}$ | $\mathbf{94.7}_{\pm1.9}$ | $\mathbf{88.0}_{\pm5.7}$ | $\mathbf{80.0}_{\pm8.6}$ | $92.0_{\pm0.0}$ | $\mathbf{77.3}_{\pm3.8}$ | $\mathbf{98.7}_{\pm1.9}$ | $\mathbf{69.3}_{\pm5.0}$ | $\mathbf{12.0}_{\pm3.7}$ |

Table 2: **Evaluation on RLBench on the single-view setup.** 3D Diffuser Actor outperforms prior state-of-the-art baselines, GNFactor and Act3D, on most tasks by a large margin.

Only the front RGB-D camera view is available. We evaluate policies by task completion success rate, which is the proportion of execution trajectories that achieve the goal conditions specified in the language instructions [15, 14].

**Baselines** All compared methods on RLBench are 3D policies that use depth and camera parameters during featurization of the RGB-D input. We compare against the following: C2F-ARM-BC [13] and PerAct [14] that voxelize the 3D workspace, Hiveformer [27] that featurizes XYZ coordinates aligned with the 2D RGB frames, PolarNet [68] that featurizes a scene 3D point cloud, GNFactor [67] that uses a single RGB-D view and is trained to complete the 3D feature volume, RVT [16] and Act3D [15], that are the previous SOTA methods on RLBench. We report results for RVT, PolarNet and GNFactor based on their respective papers. Results for CF2-ARM-BC and PerAct are presented as reported in [16]. Results for Hiveformer are copied from [68]. We retrained Act3D on the multi-view setup using the publicly available code, because we found some differences on the train and test splits used in the original paper. We also trained Act3D on the single-view setup to use as additional baseline for the single-view setup, alongside GNFactor. 3D Diffuser Actor and all baselines are trained on the same set of keyposes extracted from expert demonstrations [57].

We show quantitative results for the multi-view setup in Table 1 and for single-view setup in Table 2. On multi-view, 3D Diffuser Actor achieves an average 81.3% success rate among all 18 tasks, an absolute improvement of +18.1% over Act3D, the previous state-of-the-art. In particular, 3D Diffuser Actor achieves big leaps on long-horizon high-precision tasks with multiple modes, such as *stack blocks*, *stack cups* and *place cups*, which most baselines fail to complete. All baselines use classification or regression losses, our model is the only one to use diffusion for action prediction.

On single-view, 3D Diffuser Actor outperforms GNFactor by +46.7% and Act3D by +13.1%. Surprisingly, Act3D also outperforms GNFactor by a big margin. This suggests that the choice of 3D scene representation is more crucial than 3D feature completion. Since Act3D has a similar 3D scene tokenization as 3D Diffuser Actor, this shows the importance of diffusion over alternatives to handle multimodality in prediction, specifically over sampling based 3D action maps and rotation regression.

**Ablations** We consider the following ablative versions of our model: **1.** 2D Diffuser Actor, our implementation of 2D Diffusion Policy of [6]. We remove the 3D scene encoding from 3D Diffuser Actor and instead use per-image 2D representations by average-pooling features

within each view. We add learnable embeddings to distinguish between different views and fuse them with action estimates, as done in [6]. See Fig. 6 in the Appendix for more details. **2. 3D Diffuser Actor w/o RelAttn.**, an ablative version that uses absolute (non-relative) attentions.

We show ablative results in Table 3. 3D Diffuser Actor largely outperforms its 2D counterpart, 2D Diffuser Actor. This shows the importance of 3D scene representations in performance. 3D Diffuser Actor with absolute 3D attentions (3D Diffuser Actor w/o RelAttn.) is worse than 3D Diffuser Actor with relative 3D attentions. This shows that translation equivariance through relative attentions is important for generalization. Despite that, this ablative version already outperforms all prior arts in Table 1, proving the effectiveness of marrying 3D representations and diffusion policies.

|  | Avg. Success. |
| --- | --- |
| 2D Diffuser Actor | 47.0 |
| 3D Diffuser Actor w/o RelAttn. | 71.3 |
| 3D Diffuser Actor (ours) | **81.3** |

Table 3: **Ablation study.** Our model significantly outperforms its counterparts that do not use 3D scene representations or translation-equivariant 3D relative attention.

## 4.2 Evaluation on CALVIN

The CALVIN benchmark is built on top of the PyBullet [69] simulator and involves a Franka Panda Robot arm that manipulates the scene. CALVIN consists of 34 tasks and 4 different environments (A, B, C and D). All environments are equipped with a desk, a sliding door, a drawer, a button that turns on/off an LED, a switch that controls a lightbulb and three different colored blocks (red, blue and pink). These environments differ from each other in the texture of the desk and positions of the objects. CALVIN provides 24 hours of tele-operated unstructured play data, 35% of which are annotated with language descriptions. Each instruction chain includes five language instructions that need to be executed sequentially. We evaluate on the so called *zero-shot generalization setup*, where models are trained in environments A, B and C and tested in D. We report the success rate and the average number of completed sequential tasks, following previous works [70, 58]. No motion planner is available in CALVIN so all models need to predict robot pose trajectories. We found beneficial to employ more layers of language attention, as language understanding on CALVIN is more challenging. See Fig. 4 in the Appendix for more details.

**Baselines** All methods tested so far in CALVIN are 2D policies, that do not use depth or camera extrinsics. We compare against the hierarchical 2D policies of MCIL [71], HULC [70] and SuSIE [45] which predict latent features or images of subgoals given a language instruction, which they feed into lower-level subgoal-conditioned policies. They can train the low-level policy on all data available in CALVIN as opposed to the language annotated subset only. We compare against large scale 2D transformer policies of RT-1 [49], RoboFlamingo [72] and GR-1 [58] which pretrain on large amounts of interaction or observational (video alone) data. We report results for HULC, RoboFlamingo, SuSIE and GR-1 from the respective papers. Results from MCIL are borrowed from [70]. Results from RT-1 are copied from [58].

We also compare against 3D Diffusion Policy [22] and ChainedDiffuser [21], which we train both on the language annotated training set. We evaluate 3D Diffusion Policy, ChainedDiffuser and 3D Diffuser Actor with the final checkpoints across 3 seeds. We report both the mean and standard deviation of evaluation results. We devise an algorithm to extract keyposes on CALVIN, since prior works do not use keyposes. We define keyposes as those at frames with significant motion change. Both ChainedDiffuser and 3D Diffuser Actor segment the demonstations based on the keyposes. Notably, though keyposes extracted in RLBench have clear structure since they correspond to programmatically labeled 3D waypoints [19, 57], keyposes extracted in CALVIN are instead noisy and random, as the benchmark consists of human play trajectories.

For evaluation, for each task, our model predicts at most 60 trajectories in sequence, one after the other (it conducts 60 inferences per task). Each trajectory is on average 6 action long. We did this to ensure a fair comparison to prior works that predict a maximum of 360 actions in total per task [70, 45, 72].

We show quantitative results in Table 4. 3D Diffuser Actor outperforms the state-of-the-art. ChainedDiffuser does not work well on this benchmark, as its deterministic keypose prediction module fails to predict end-effector keyposes accurately due to multimodality present in human demonstrations of CALVIN, in comparison to programmatically collected demonstrations of RLBench.

| | Train episodes | Task completed in a row | | | | | |
|---|---|---|---|---|---|---|---|
| | | 1 | 2 | 3 | 4 | 5 | Avg. Len |
| 3D Diffusion Policy [22] | Lang | $28.7_{\pm0.4}$ | $2.7_{\pm0.4}$ | $0.0_{\pm0.0}$ | $0.0_{\pm0.0}$ | $0.0_{\pm0.0}$ | $0.31_{\pm0.04}$ |
| MCIL [71] | All | 30.4 | 1.3 | 0.2 | 0.0 | 0.0 | 0.31 |
| HULC [70] | All | 41.8 | 16.5 | 5.7 | 1.9 | 1.1 | 0.67 |
| RT-1 [49] | Lang | 53.3 | 22.2 | 9.4 | 3.8 | 1.3 | 0.90 |
| ChainedDiffuser [21] (60 keyposes) | Lang | $49.9_{\pm0.01}$ | $21.1_{\pm0.01}$ | $8.0_{\pm0.01}$ | $3.5_{\pm0.0}$ | $1.5_{\pm0.0}$ | $0.84_{\pm0.02}$ |
| RoboFlamingo [72] | Lang | 82.4 | 61.9 | 46.6 | 33.1 | 23.5 | 2.48 |
| SuSIE [45] | All | 87.0 | 69.0 | 49.0 | 38.0 | 26.0 | 2.69 |
| GR-1 [58] | Lang | 85.4 | 71.2 | 59.6 | 49.7 | 40.1 | 3.06 |
| 3D Diffuser Actor (ours) | Lang | $\mathbf{93.8}_{\pm0.01}$ | $\mathbf{80.3}_{\pm0.0}$ | $\mathbf{66.2}_{\pm0.01}$ | $\mathbf{53.3}_{\pm0.02}$ | $\mathbf{41.2}_{\pm0.01}$ | $\mathbf{3.35}_{\pm0.04}$ |

Table 4: **Zero-shot long-horizon evaluation on CALVIN** on 3 random seeds.

## 4.3 Evaluation in the real world

We validate 3D Diffuser Actor in learning manipulation tasks from real-world demonstrations across 12 tasks. We use a Franka Emika robot equipped with a Azure Kinect RGB-D sensor at a front view. Images are originally captured at $1280 \times 720$ resolution and downsampled to a resolution of $256 \times 256$. During inference, we utilize the BiRRT [66] planner provided by the MoveIt! ROS package [73] to reach the predicted poses. We collect 15 demonstrations per task, most of which naturally contain noise and multiple modes of human behavior. For example, we pick one of two ducks to put in the bowl, we

| close box | put duck | insert peg into hole | insert peg into torus | put mouse | open pen |
|---|---|---|---|---|---|
| 100 | 100 | 50 | 30 | 80 | 100 |

| press stapler | put grapes | sort rectangle | stack blocks | stack cups | put block in triangle |
|---|---|---|---|---|---|
| 90 | 90 | 50 | 20 | 40 | 90 |

Table 5: **Multi-Task performance on real-world tasks.**

insert the peg into one of two holes and we put one of three grapes in the bowl. We evaluate 10 episodes for each task and report the success rate. We show quantitative results in Table 5 and video results on our project webpage. 3D Diffuser Actor effectively learns real-world manipulation from a handful of demonstrations.

**Runtime** We compare the inference time of our model against ChainedDiffuser [21] and 3D Diffusion Policy [22] on CALVIN, using an NVIDIA 2080 Ti graphic card. The inference speed of 3D Diffuser Actor, ChainedDiffuser and 3D Diffusion Policy is 600ms, 1170ms (50 for keypose detection and 1120 for trajectory optimization) and 581ms.

**Limitations** Despite its SOTA performance with large margins over existing methods, our framework currently has the following limitations: **1.** It requires camera calibration and depth information, same as all 3D policies. **2.** All tasks in RLBench and CALVIN are quasi-static. Extending our method to dynamic tasks and velocity control is a direct avenue of future work. **3.** It is on average slower than non-diffusion policies. This can be improved by recent techniques on reducing the inference steps of diffusion models.

Please, refer to our Appendix for more experiments and details: Section A.1 for robustness of 3D Diffuser Actor to depth noise, Section A.2 for discussion of failure cases, Section A.3 and A.4 for descriptions of tasks in RLBench and the real world, Section A.5 for descriptions of baselines, Section A.5.1 for implementation details of re-training 3D Diffusion Policy on CALVIN, Section A.6 for keypose discovery on CALVIN. Video results can be found in our supplementary file.

## 5 Conclusion

We present 3D Diffuser Actor, a manipulation policy that combines 3D scene representations and action diffusion. Our method sets a new state-of-the-art on RLBench and CALVIN by a large margin over existing 2D and 3D policies, and learns robot control in the real world from a handful of demonstrations. 3D Diffuser Actor builds upon recent progress on 3D tokenized scene representations for robotics and on generative models and shows how their combination is a powerful learning from demonstration method. Our future work includes learning 3D Diffuser Actor policies from suboptimal demonstrations and scaling up training data in simulation and in the real world.

# 6 Acknowledgements

This work is supported by Sony AI, NSF award No 1849287, DARPA Machine Common Sense, an Amazon faculty award, and an NSF CAREER award. The authors would like to thank Moritz Reuss for useful training tips on CALVIN; Zhou Xian for help with the real-robot experiments; Brian Yang for discussions, comments and efforts in the early development of this paper.

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

# Appendix

# A  Additional Experimental Results and Details

## A.1  Robustness to noisy depth information on RLBench

We evaluate the robustness of 3D Diffuser Actor to noisy depth information. Our results are presented in Table 6. We adopt the single-view setup on RLBench, reporting success rates on 10 tasks using the *front* camera view. We add Gaussian noise with a variance of $0.01$ and $0.03$ to the 3D location of each pixel. We evaluate the final checkpoints across 3 seeds with 25 episodes for each task. With mild perturbation, 3D Diffuser Actor can still predict actions precisely, achieving an average success rate of $72.4\%$. We only observed $6\%$ absolute performance drop. With strong perturbation, 3D Diffuser Actor obtains an average success rate of $50.1\%$. Even though 3D Diffuser Actor was trained with clean depth maps, it performs reasonably well with noisy depth information.

| $\sigma$ | Avg. Success. | close jar | open drawer | sweep to dustpan | meat off grill | turn tap | slide block | put in drawer | drag stick | push buttons | stack blocks |
|---|---|---|---|---|---|---|---|---|---|---|---|
| 0 (clean) | **78.4** | **82.7**$_{\pm1.9}$ | **89.3**$_{\pm7.5}$ | **94.7**$_{\pm1.9}$ | **88.0**$_{\pm5.7}$ | 80.0$_{\pm8.6}$ | 92.0$_{\pm0.0}$ | **77.3**$_{\pm3.8}$ | **98.7**$_{\pm1.9}$ | 69.3$_{\pm5.0}$ | **12.0**$_{\pm3.7}$ |
| 0.01 | 72.4 | 80.0$_{\pm3.3}$ | 81.3$_{\pm1.9}$ | 68.0$_{\pm3.3}$ | 85.3$_{\pm5.0}$ | **82.7**$_{\pm7.5}$ | 92.0$_{\pm0.0}$ | 65.3$_{\pm3.8}$ | 93.3$_{\pm3.8}$ | **70.7**$_{\pm1.9}$ | 5.3$_{\pm1.9}$ |
| 0.03 | 50.1 | 41.3$_{\pm6.8}$ | 25.3$_{\pm10.0}$ | 45.3$_{\pm3.8}$ | 74.7$_{\pm3.8}$ | 89.3$_{\pm3.8}$ | 89.3$_{\pm3.8}$ | 18.7$_{\pm1.9}$ | 48.0$_{\pm3.3}$ | 68.0$_{\pm5.7}$ | 1.3$_{\pm1.9}$ |

$\sigma = 0$ (clean) $\quad\quad\quad$ $\sigma = 0.01$ $\quad\quad\quad$ $\sigma = 0.03$

Table 6: **Multi-Task performance on noisy depth information. Top:** We evaluate the robustness of 3D Diffuser Actor to noisy depth information. We adopt the single-view setup in RLBench, reporting success rates on 10 tasks using the *front* camera view. We add Gaussian noise with a variance of $0.01$ and $0.03$ to the 3D location of each pixel. We evaluate the final checkpoints across 3 seeds with 25 episodes for each task. 3D Diffuser Actor performs reasonably well with noisy depth information. **Bottom:** Visualization of the 3D location of each pixel with different levels of noise.

## A.2  Failure cases on RLBench

We analyze the failure modes of 3D Diffuser Actor on RLBench. We categorize the failure cases into 4 types: 1) pose precision, where predicted end-effector poses are too imprecise to satisfy the success condition, 2) instruction understanding, where the policy fails to understand the language instruction to grasp the target object, 3) long-horizon task completion, where the policy fails to complete an intermediate task of a long-horizon task, and 4) path planning, where the motion planner fails to find an optimal path to reach the target end-effector pose.

We conduct the experiment with single-view setup on RLBench, aggregating the failure cases among 10 manipulation tasks. As shown in Fig. 2, the major failure mode is imprecise prediction of end-effector poses. This failure mode often occurs in manipulation tasks that require high precision of predicted poses, such as *stack blocks*, *open drawer* and *turn tap*. Confusion of language instruction is another major failure mode. This mode often occurs in the scene with multiple objects, where the policy fails to grasp the target object specified in the instruction, such as *stack blocks*, *close jar* and *push buttons*. On RLBench, we deploy a sample-based motion planner to find the path to reach the keypose. The motion planner could sometimes fail to find the path. Lastly, 3D Diffuser Actor might fail to complete an intermediate task, such as *open drawer* of *put item in drawer* task.

In the following, we discuss ideas on how to mitigate each failure case:

- Imprecise pose prediction: e.g., the model reaches the right object but fails to grasp it due to last centimeter errors (misses while executing fine motor control). We expect that aggregating datasets and training the policy on a wider and more diverse set of manipulation tasks would benefit learning finer control.
- Language understanding: e.g., the model misunderstands the language instruction and grasps the wrong object. Equiping our model with perception and language understanding

modules from modern VLMs, such as explicitly detecting the target objects through a VLM pre-trained on internet-scale data, would drastically improve the cognitive capabilities of our model.

- Intermediate tasks: e.g., failing to open the drawer, when the task is to put something inside the drawer. We believe that decomposing a task into simpler ones through high-level planning (e.g., using LLMs) and training reward models for independent subtasks will boost our model's generalization to longer-horizon tasks.

- Planner failures: we can predict the full trajectory up to the next keypose, instead of relying to a planner, as we do on CALVIN. We didn't do it on RLBench since the amount of planning errors is small.

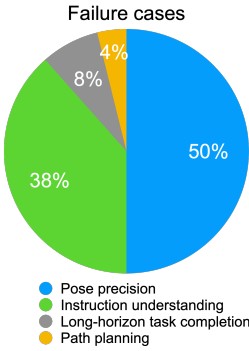

Failure cases

- Pose precision
- Instruction understanding
- Long-horizon task completion
- Path planning

Figure 2: **Failure cases on RLBench on the setup of GNFactor.** We categorize the failure cases into 4 types: **1)** precise pose prediction, where predicted end-effector poses are too imprecise to satisfy the success condition, **2)** instruction understanding, where the policy fails to understand the language instruction to grasp the target object, **3)** long-horizon task completion, where the policy fails to complete an intermediate task of a long-horizon task, and **4)** path planning, where the motion planner fails to find an optimal path to reach the target end-effector pose. Imprecise action prediction and confusion of language instruction are two major failure modes on RLBench.

## A.3 RLBench tasks under multi-view setup

We provide an explanation of the RLBench tasks and their success conditions under the multi-view setup for self-completeness. All tasks vary the object pose, appearance and semantics, which are not described in the descriptions below. For more details, please refer to the PerAct paper [14].

1. Open a drawer: The cabinet has three drawers (top, middle and bottom). The agent is successful if the target drawer is opened. The task on average involves three keyposes.

2. Slide a block to a colored zone: There is one block and four zones with different colors (red, blue, pink, and yellow). The end-effector must push the block to the zone with the specified color. On average, the task involves approximately 4.7 keyposes

3. Sweep the dust into a dustpan: There are two dustpans of different sizes (short and tall). The agent needs to sweep the dirt into the specified dustpan. The task on average involves 4.6 keyposes.

4. Take the meat off the grill frame: There is chicken leg or steak. The agent needs to take the meat off the grill frame and put it on the side. The task involves 5 keyposes.

5. Turn on the water tap: The water tap has two sides of handle. The agent needs to rotate the specified handle $90°$. The task involves 2 keyposes.

6. Put a block in the drawer: The cabinet has three drawers (top, middle and bottom). There is a block on the cabinet. The agent needs to open and put the block in the target drawer. The task on average involves 12 keyposes.

7. Close a jar: There are two colored jars. The jar colors are sampled from a set of 20 colors. The agent needs to pick up the lid and screw it in the jar with the specified color. The task involves six keyposes.

8. Drag a block with the stick: There is a block, a stick and four colored zones. The zone colors are sampled from a set of 20 colors. The agent is successful if the block is dragged to the specified colored zone with the stick. The task involves six keyposes.

9. Stack blocks: There are 8 colored blocks and 1 green platform. Each four of the 8 blocks share the same color, while differ from the other. The block colors are sampled from a set of 20 colors. The agent needs to stack N blocks of the specified color on the platform. The task involves 14.6 keyposes.

10. Screw a light bulb: There are 2 light bulbs, 2 holders, and 1 lamp stand. The holder colors are sampled from a set of 20 colors. The agent needs to pick up and screw the light bulb in the specified holder. The task involves 7 keyposes.

11. Put the cash in a safe: There is a stack of cash and a safe. The safe has three layers (top, middle and bottom). The agent needs to pick up the cast and put it in the specified layer of the safe. The task involves 5 keyposes.

12. Place a wine bottle on the rack: There is a bottle of wine and a wooden rack. The rack has three slots (left, middle and right). The agent needs to pick up and place the wine at the specified location of the wooden rack. The task involves 5 keyposes.

13. Put groceries in the cardboard: There are 9 YCB objects and a cupboard. The agent needs to grab the specified object and place it in the cupboard. The task involves 5 keyposes.

14. Put a block in the shape sorter: There are 5 blocks of different shapes and a sorter with the corresponding slots. The agent needs to pick up the block with the specified shape and insert it into the slot with the same shape. The task involves 5 keyposes.

15. Push a button: There are 3 buttons, whose colors are sampled from a set of 20 colors. The agent needs to push the colored buttons in the specified sequence. The task involves 3.8 keyposes.

16. Insert a peg: There is 1 square, and 1 spoke platform with three colored spoke. The spoke colors are sampled from a set of 20 colors. The agent needs to pick up the square and put it onto the spoke with the specified color. The task involves 5 keyposes.

17. Stack cups: There are 3 cups. The cup colors are sampled from a set of 20 colors. The agent needs to stack all the other cups on the specified one. The task involves 10 keyposes.

18. Hang cups on the rack: There are 3 mugs and a mug rack. The agent needs to pick up N mugs and place them onto the rack. The task involves 11.5 keyposes.

## A.4   Real-world tasks

We explain the the real-world tasks and their success conditions in more detail. All tasks take place in a cluttered scene with distractors (random objects that do not participate in the task) which are not mentioned in the descriptions below.

1. Close a box: The end-effector needs to move and hit the lid of an open box so that it closes. The agent is successful if the box closes. The task involves two keyposes.

2. Put a duck in a bowl: There are two toy ducks and two bowls. One of the ducks have to be placed in one of the bowls. The task involves four keyposes.

3. Insert a peg vertically into the hole: The agent needs to detect and grasp a peg, then insert it into a hole that is placed on the ground. The task involves four keyposes.

4. Insert a peg horizontally into the torus: The agent needs to detect and grasp a peg, then insert it into a torus that is placed vertically to the ground. The task involves four keyposes.

5. Put a computer mouse on the pad: There two computer mice and one mousepad. The agent needs to pick one mouse and place it on the pad. The task involves four keyposes.

6. Open the pen: The agent needs to detect a pen that is attached vertically to the table, grasp its lid and pull it to open the pen. The task involves three keyposes.

7. Press the stapler: The agent needs to reach and press a stapler. The task involves two keyposes.

8. Put grapes in the bowl: The scene contains three vines of grapes of different color and one bowl. The agent needs to pick one vine and place it in the bowl. The task involves four keyposes.

9. Sort the rectangle: Between two rectangle cubes there is space for one more. The task comprises detecting the rectangle to be moved and placing it between the others. It involves four keyposes.

10. Stack blocks with the same shape: The scene contains of several blacks, some of which have rectangular and some cylindrical shape. The task is to pick the same-shape blocks and stack them on top of the first one. It involves eight keyposes.

11. Stack cups: The scene contains three cups of different colors. The agent needs to successfully stack them in any order. The task involves eight keyposes.

12. Put block in a triangle on the plate: The agent needs to detect three blocks of the same color and place them inside a plate to form an equilateral triangle. The task involves 12 keyposes.

The above tasks examine different generalization capabilities of 3D Diffuser Actor, for example multimodality in the solution space (5, 8), order of execution (10, 11, 12), precision (3, 4, 6) and high noise/variance in keyposes (1).

### A.5 Additional details on baselines

We provide more details of the baselines used in this paper. For RLBench, we consider the following baselines:

1. C2F-ARM-BC [13], a 3D policy that iteratively voxelizes RGB-D images and predicts actions in a coarse-to-fine manner. Q-values are estimated within each voxel and the translation action is determined by the centroid of the voxel with the maximal Q-values.

2. PerAct [14], a 3D policy that voxelizes the workspace and detects the next voxel action through global self-attention.

3. Hiveformer [27], a 3D policy that enables attention between features of different history time steps.

4. PolarNet [68], a 3D policy that computes dense point representations for the robot workspace using a PointNext backbone [74].

5. RVT [16], a 3D policy that deploys a multi-view transformer to predict actions and fuses those across views by back-projecting to 3D.

6. Act3D [15], a 3D policy that featurizes the robot's 3D workspace using coarse-to-fine sampling and featurization. We observed that Act3D does not follow the same setup as PerAct on RLBench. Specifically, Act3D uses different 1) 3D object models, 2) success conditions, 3) training/test episodes and 4) maximum numbers of keyposes during evaluation. For fair comparison, we retrain and test Act3D on the same setup.

7. GNFactor [67], a 3D policy that co-optimizes a neural field for reconstructing the 3D voxels of the input scene and a PerAct module for predicting actions based on voxel representations.

We consider the following baselines for CALVIN:

1. MCIL [71], a multi-modal goal-conditioned 2D policy that maps three types of goals–goal images, language instructions and task labels–to a shared latent feature space, and conditions on such latent goals to predict actions.

2. HULC [70], a 2D policy that uses a variational autoencoder to sample a latent plan based on the current observation and task description, then conditions on this latent to predict actions.

3. RT-1 [49], a 2D transformer-based policy that encodes the image and language into a sequence of tokens and employs a Transformer-based architecture that contextualizes these tokens and predicts the arm movement or terminates the episode.

4. RoboFlamingo [72], a 2D policy that adapts existing vision-language models, which are pre-trained for solving vision and language tasks, to robot control. It uses frozen vision and language foundational models and learns a cross-attention between language and visual features, as well as a recurrent policy that predicts the low-level actions conditioned on the language latents.

5. SuSIE [45], a 2D policy that deploys an large-scale pre-trained image generative model [75] to synthesize visual subgoals based on the current observation and language instruction. Actions are then predicted by a low-level goal-conditioned 2D diffusion policy that models inverse dynamics between the current observation and the predicted subgoal image.

6. GR-1 [58], a 2D policy that first pre-trains an autoregressive Transformer on next frame prediction, using a large-scale video corpus without action annotations. Each video frame is encoded into an 1d vector by average-pooling its visual features. Then, the same architecture is fine-tuned in-domain to predict both actions and future observations.

7. ChainedDiffuser [21], a combination of Act3D [15] that predicts end-effector poses at key frames and a 3D trajectory predictor that generates intermediate trajectories to reach target end-effector poses.

8. 3D Diffusion Policy [22], that encodes sparsely sampled point cloud into 3D representations using a PointNeXt [74] encoder. The 3D representations are average pooled into a 1D feature vector. Then, a UNet conditions on the holistic 3D scene representations and predicts the robot end-effector poses. Notably, in its original implementation, 3D Diffusion Policy is only applied to single-task experimental setup and does not condition on language instructions. For fair comparison, we update the architecture to enable language conditioning. See Section A.5.1 for more details.

### A.5.1 Re-training of 3D Diffusion Policy on CALVIN

We compare 3D Diffuser Actor against 3D Diffusion Policy on CALVIN. However, in its original implementation, the 3D Diffusion Policy does not condition on language instructions and is applied only in a single-task setup. For a fair comparison, we enabled language conditioning in the 3D Diffusion Policy by updating its architecture with additional point-cloud-to-language cross-attention layers. We used a language encoder similar to 3D Diffuser Actor to encode language instructions into latent embeddings and applied three cross-attention layers among average pooled point-cloud features and all language tokens. To ensure the maximal performance, we do not use the simplified backbone of DP3 (*Simple DP3*), but the original DP3 architecture.

We adopt the common setup in CALVIN, using both the *front* and *wrist* camera view. Based on the suggestion of the paper [22], we crop a $160 \times 160$ and $68 \times 68$ bounding box from the depth map of the *front* and *wrist* camera, sampling $1024$ points within each bounding box. We train 3D Diffusion Policy with a batch size of 5400 and a total epoch of 3000. Otherwise, we use the default hyper-parameters of 3D Diffusion Policy. We set the horizon of action prediction to 4, the number of executed actions to 3, and the number observed timesteps to 2. During inference, we allow the model to predict 360 times, resulting in a maximum temporal horizon of 1080 actions.

### A.6 Keypose discovery

For RLBench we use the heuristics from [57, 13]: a pose is a keypose if (1) the end-effector state changes (grasp or release) or (2) the velocity's magnitude approaches zero (often at pre-grasp poses or a new phase of a task). For our real-world experiments we maintain the above heuristics and record pre-grasp poses as well as the poses at the beginning of each phase of a task, e.g., when the end-effector is right above an object of interest. We report the number of keyposes per real-world task in this Appendix (Section A.4). Lastly, for CALVIN we adapt the above heuristics to devise a more robust algorithm to discover keyposes. Specifically, we track end-effector state changes and significant changes of motion, i.e. both velocity and acceleration. For reference, we include our Python code for discovering keyposes in CALVIN here:

```python
"""Utility functions for computing keyposes."""
import numpy as np
from scipy.signal import argrelextrem

def motion_changed(trajectories, buffer_size):
    """Select keyposes where motion changes significantly. The chosen
    poses shall be sparse.

    Args:
        trajectories: a list of 1D array
        buffer_size: an integer indicates the
            minimum distance of waypoints

    Returns:
        key_poses: a list of integers indicates
            the time steps when motion of the end
            effector changes significantly.
    """
    # compute velocity
    trajectories = np.stack(
```

```python
20          [trajectories[0]] + trajectories, axis=0
21      )
22      velocities = (
23          trajectories[1:] - trajectories[:-1]
24      )
25      # compute acceleration
26      velocities = np.concatenate(
27          [velocities, [velocities[-1]]],
28          axis=0
29      )
30      accelerations = velocities[1:] - velocities[:-1]
31      # compute the magnitude of acceleration
32      A = np.linalg.norm(
33          accelerations[:, :3], axis=-1
34      )
35      # local maximas of acceleration indicates
36      # significant motion change of the end effector
37      local_max_A = argrelextrema(A, np.greater)[0]
38
39      # consider the top 20% of local maximas
40      K = int(A.shape[0] * 0.2)
41      topK = np.sort(A)[::-1][K]
42      large_A = A[local_max_A] >= topK
43      local_max_A = local_max_A[large_A].tolist()
44
45      # select waypoints sparsely
46      key_poses = [local_max_A.pop(0)]
47      for i in local_max_A:
48          if i - key_poses[-1] >= buffer_size:
49              key_poses.append(i)
50
51      return key_poses
52
53  def gripper_state_changed(trajectories):
54      """Select keyposes where the end-effector
55      opens/closes.
56
57      Args:
58          trajectories: a list of 1D array
59
60      Returns:
61          key_poses: a list of integers indicates
62              the time steps when the end-effector
63              opens/closes.
64      """
65      trajectories = np.stack(
66          [trajectories[0]] + trajectories, axis=0
67      )
68      openess = trajectories[:, -1]
69      changed = openess[:-1] != openess[1:]
70      key_poses = np.where(changed)[0].tolist()
71
72      return key_poses
73
74  def keypoint_discovery(trajectories, buffer_size=5):
75      """Select keyposes where motion changes significantly. The chosen
76      poses shall be sparse.
77
78      Args:
79          trajectories: a list of 1D array
80          buffer_size: an integer indicates the
81              minimum distance of waypoints
82
83      Returns:
84          key_poses: an Integer array indicates the
```

```
84            indices of keyposes
85     """
86     motion_changed = motion_changed(
87         trajectories, buffer_size
88     )
89
90     gripper_changed = (
91         gripper_state_changed(trajectories)
92     )
93     one_frame_before_gripper_changed = [
94         i - 1 for i in gripper_changed if i > 1
95     ]
96
97     last_frame = [len(trajectories) - 1]
98
99     key_pose_inds = (
100         moition_changed +
101         gripper_changed.tolist() +
102         one_frame_before_gripper_changed.tolist() +
103         last_frame
104     )
105     key_pose_inds = np.unique(key_pose_inds)
106
107     return key_pose_inds
```

### A.7 Comparison between 3D Diffuser Actor and 3D Diffusion Policy under single-task setup

We test 3D Diffusion Policy [22] to our 3D Diffuser Actor on the stack-cup task of RLBench. Under single-task setup, we do not need to condition on language instruction, which matches the original setup of DP3. We trained both methods on 100 episodes and tested on 25 unseen episodes. As shown in Table 7, 3D Diffuser Actor achieves a success rate of 80%, and 3D Diffusion Policy achieves a success rate of 8%. We believe this experiment clearly highlights the importance of our tokenized 3D scene and action trans-

|  | stack cups |
|---|---|
| 3D Diffuser Actor | 80 |
| 3D Diffusion Policy [22] | 8 |

Table 7: **Single-Task performance on stack-cup task of RLBench.**

former, missing from 3D Diffusion Policy, which encodes a whole 3D point cloud into a single embedding vector.

## B Additional Method Details

### B.1 Architectural differences between our model and baselines

We describe the architectural differences between 3D Diffuser Actor and other 3D policies in Figure 3. We compare 3D Diffuser Actor with Act3D [15], PerAct [14], and 3D Diffusion Policy [22]. PerAct encodes RGB-D images into voxel representations, while 3D Diffusion Policy average pools point cloud representations into a holistic 1D feature vector. Act3D encodes input images with a 2D feature extractor and lifts 2D feature maps to 3D. Act3D determines end-effector poses by iterating a coarse-to-fine classification procedure to identify the XYZ location of the ghost point with the highest classification score. Although our model adopts a similar 3D scene encoder as Act3D, 3D Diffuser Actor uses a relative 3D transformer to denoise end-effector poses.

### B.2 The formulation of relative attention

To ensure translation invariance in the denoising transformer, we use the rotary positional embeddings [18] to encode relative positional information in attention layers. The attention between query $\mathbf{q}_i$, key $\mathbf{k}_j$ and value $\mathbf{v}_j$ is written as:

$$\text{Attention}(\mathbf{q}_i, \mathbf{k}_j, \mathbf{v}_j) = \frac{\exp e_{i,j}}{\sum_l \exp e_{i,l}} \mathbf{v}_j \tag{1}$$

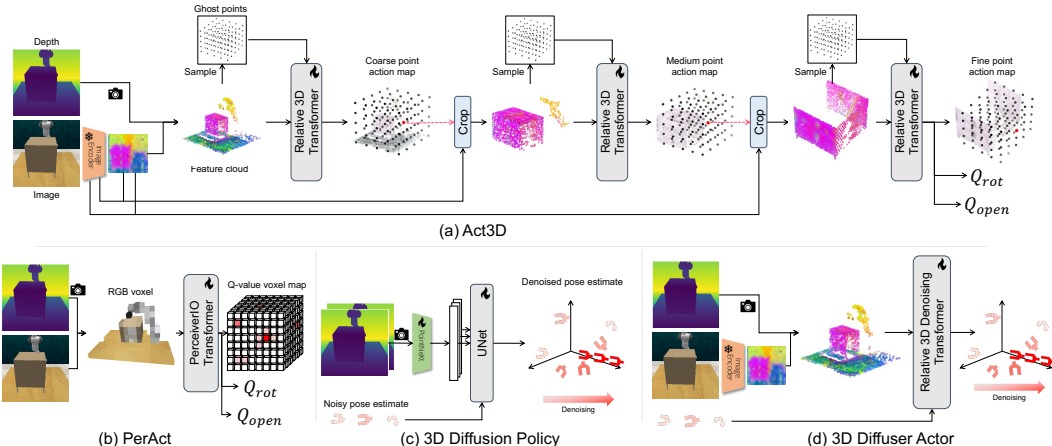

(a) Act3D

(b) PerAct        (c) 3D Diffusion Policy        (d) 3D Diffuser Actor

Figure 3: **Comparison of 3D Diffuser Actor and other methods**. **(a)** Act3D [15] encodes input images into with a 2D feature extractor, lifts 2D feature maps to 3D, samples ghost points within the scene, featurizes ghost points with relative cross attention to the 3D feature cloud, and classifies each ghost point. The position is determined by the XYZ location of the ghost point with the highest score. Act3D then crops the scene with predicted end-effector position, and iterates the same classification procedure. The rotation and openess are predicted with learnable query embeddings. **(b)** PerAct [14] voxelizes input images and uses a Perceiver Transformer [56] to predict the end-effector pose. The position is determined by the XYZ location of the voxel with the highest Q-value. The rotation and openess are predicted from the max-pooled features with MLPs. **(c)** 3D Diffusion Policy [22] encodes 3D point cloud into 1D feature vectors, followed by a A UNet that conditions on point-cloud features and denoises end-effector poses. **(d)** 3D Diffuser Actor uses a similar 3D scene encoder as Act3D. A relative 3D transformer conditions on 3D feature cloud to denoise end-effector poses.

where $e_{i,j} = \mathbf{q}_i^T \mathbf{M}(\mathbf{p}_j - \mathbf{p}_i)\mathbf{k}_j$, $\mathbf{p}_i$ / $\mathbf{p}_j$ denote the positions of the query / key, and $\mathbf{M}$ is a matrix function which depends only on the relative positions of points $\mathbf{p}_i$ and $\mathbf{p}_j$.

### B.3 Detailed Model Diagram of 3D Diffuser Actor

We present a more detailed architecture diagram of our 3D Diffuser Actor in Figure 4a. We also show a variant of 3D Diffuser Actor with enhanced language conditioning in Figure 4b, which achieves SOTA results on CALVIN.

The inputs to our network are i) a stream of RGB-D views; ii) a language instruction; iii) proprioception in the form of current end-effector's poses; iv) the current noisy estimates of position and rotation; v) the diffusion step $i$. The images are encoded into visual tokens using a pretrained 2D backbone. The depth values are used to "lift" the multi-view tokens into a 3D feature cloud. The language is encoded into feature tokens using a language backbone. The proprioception is represented as learnable tokens with known 3D locations in the scene. The noisy estimates are fed to linear layers that map them to high-dimensional vectors. The denoising step is fed to an MLP.

The visual tokens cross-attend to the language tokens and get residually updated. The proprioception tokens attend to the visual tokens to contextualize with the scene information. We subsample a number of visual tokens using Farthest Point Sampling (FPS) in order to decrease the computational requirements. The sampled visual tokens, proprioception tokens and noisy position/rotation tokens attend to each other. We modulate the attention using adaptive layer normalization and FiLM [76]. Lastly, the contextualized noisy estimates are fed to MLP to predict the error terms as well as the end-effector's state (open/close).

### B.4 Detailed Model Diagram of Act3D

To make the paper self-contained, we present a detailed architecture diagram of Act3D in Figure 5. Act3D first encodes posed RGB-D images into 3D scene tokens and samples ghost points uniformly from a fixed-radius sphere, centered at the current end-effector position. Act3D initializes a learnable

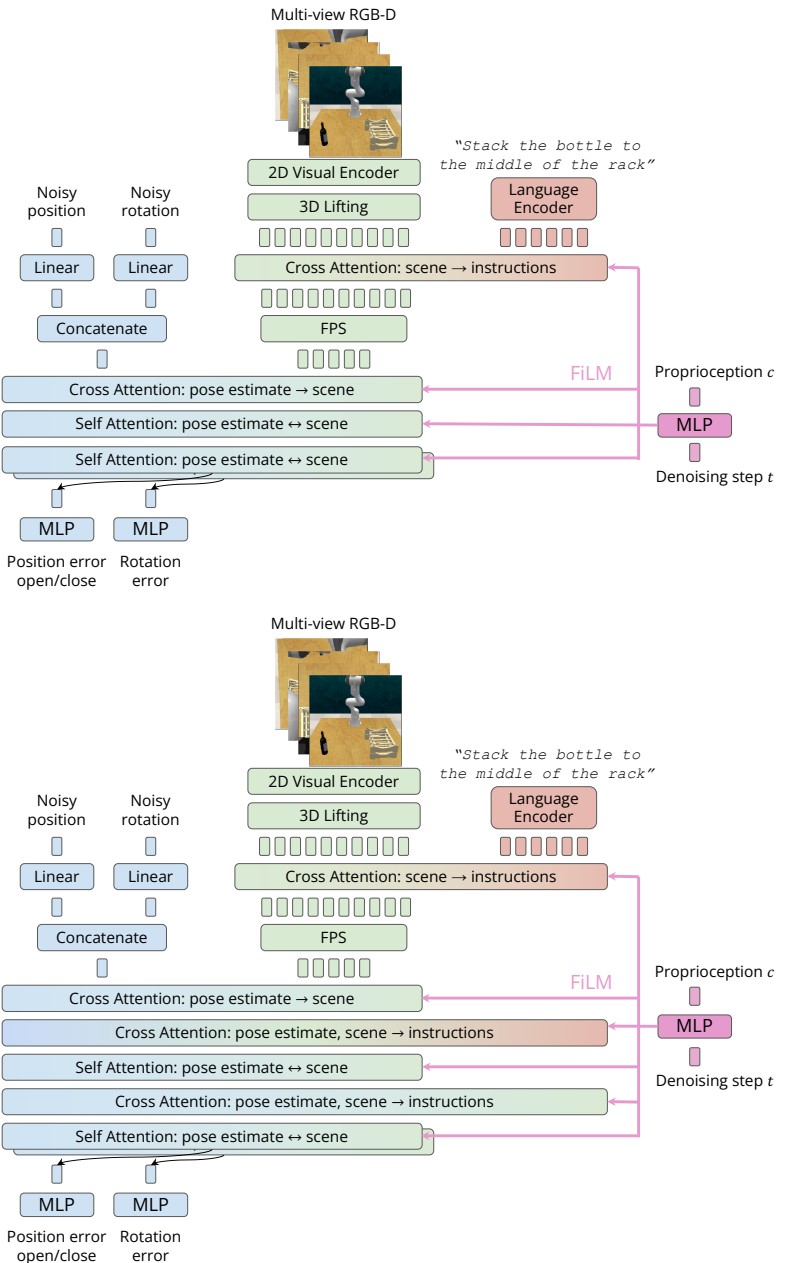

Figure 4: **3D Diffuser Actor architecture** in more detail. **Top**: Standard version: the encoded inputs are fed to attention layers that predict the position and rotation error for each trajectory timestep. The language information is fused to the visual stream by allowing the encoded visual feature tokens to attend to language feature tokens. There are two different attention and output heads for position and rotation error respectively. **Bottom**: version with enhanced language conditioning: cross-attention layers from visual and pose estimate tokens to language tokens are interleaved between pose estimate-visual token attention layers.

query and the ghost points with learnable latent embeddings, which are contexturalized by 3D scene tokens using 3D relative attention (Eqn. 1). Act3D measures the feature similarity of each ghost point and the learnable query to classify if a ghost point is close to the position of target end-effector pose, which is determined by the XYZ location of the ghost point with the highest score. Act3D then crops

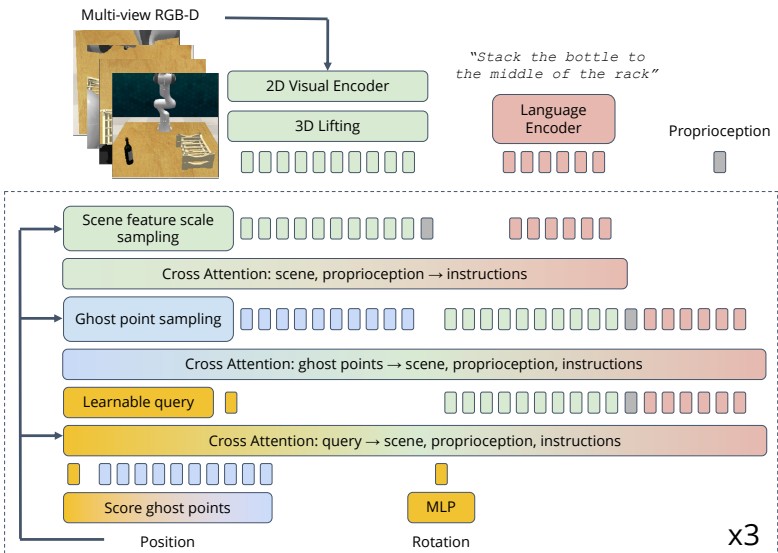

Figure 5: **Act3D architecture in more detail.** Ghost points and encoded scene feature inputs are sampled based on the current end-effector position. The 3D location of ghost points are even sampled from a fixed-radius sphere, which is centered at the current end-effector position.

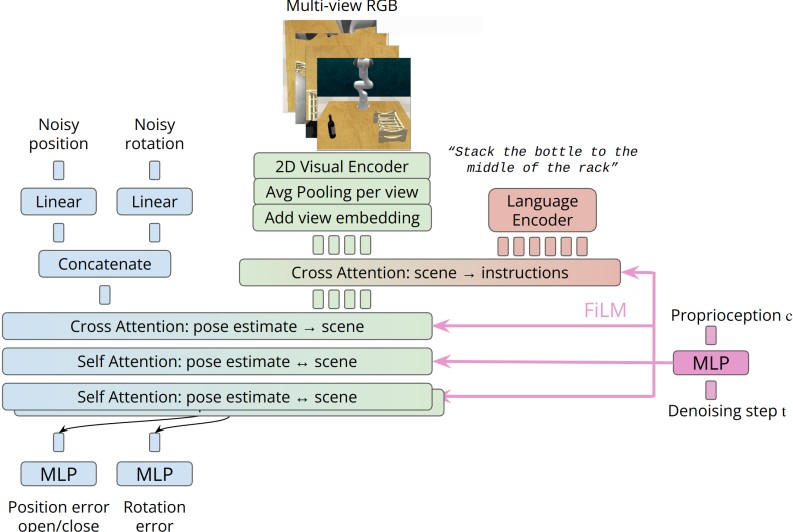

Figure 6: **Architecture of 2D Diffuser Actor, our re-implementation of 2D Diffusion Policy [6].**

the scene with estimated end-effector position, and iterates the same classification procedure. The rotation and openess are predicted with learnable query embeddings.

### B.5 Hyper-parameters for experiments

Table 8 summarizes the hyper-parameters used for training/evaluating our model. On CALVIN we observed that our model overfits the training data, resulting in lower test performance. We use higher weight_decay and shorter total_epoch on CALVIN compared to RLBench.

### B.6 Denoising Diffusion Probabilistic Models

For self-completeness, we provide a formulation of denoising diffusion process. Please check [60] for more details. A diffusion model learns to model a probability distribution $p(x)$ by inverting a process

|  | Multi-view RLBench | Single-view RLBench | CALVIN |
|---|---|---|---|
| **Model** | | | |
| image_size | 256 | 256 | 200 |
| embedding_dim | 120 | 120 | 192 |
| camera_views | 4 | 1 | 2 |
| FPS : % of sampled tokens | 20% | 20% | 33% |
| diffusion_timestep | 100 | 100 | 25 |
| noise_scheduler : position | scaled_linear | scaled_linear | scaled_linear |
| noise_scheduler : rotation | squaredcos | squaredcos | squaredcos |
| action_space | absolute pose | absolute pose | relative displacement |
| # of parameters | 3.6M | 3.6M | 9.6M |
| **Training** | | | |
| batch_size | 240 | 240 | 5400 |
| learning_rate | $1e^{-4}$ | $1e^{-4}$ | $3e^{-4}$ |
| weight_decay | $5e^{-4}$ | $5e^{-4}$ | $5e^{-3}$ |
| total_epochs | $1.6e^4$ | $8e^5$ | 90 |
| optimizer | Adam | Adam | Adam |
| loss weight : $w_1$ | 30 | 30 | 30 |
| loss weight : $w_2$ | 10 | 10 | 10 |
| **Evaluation** | | | |
| maximal # of keyposes | 25 | 25 | 60 |

Table 8: **Hyper-parameters of our experiments.** We list the hyper-parameters used for training/evaluating our model on RLBench and CALVIN simulated benchmarks. On RLBench we conduct experiments under multi-view and single-view setup.

that gradually adds noise to a sample $x$. For us, $x$ represents a sequence of 3D translations and 3D rotations for the robot's end-effector. The diffusion process is associated with a variance schedule $\{\beta^i \in (0,1)\}_{i=1}^N$, which defines how much noise is added at each time step. The noisy version of sample $x$ at time $i$ can then be written as $x^i = \sqrt{\bar{\alpha}^i}x + \sqrt{1-\bar{\alpha}^i}\epsilon$ where $\epsilon \sim \mathcal{N}(\mathbf{0},\mathbf{1})$, is a sample from a Gaussian distribution (with the same dimensionality as $x$), $\alpha^i = 1 - \beta^i$, and $\bar{\alpha}^i = \prod_{j=1}^i \alpha^j$. The denoising process is modeled by a neural network $\hat{\epsilon} = \epsilon_\theta(x^i; i)$ that takes as input the noisy sample $x^i$ and the noise level $i$ and tries to predict the noise component $\epsilon$.

Diffusion models can be easily extended to draw samples from a distribution $p(x|\mathbf{c})$ conditioned on input $\mathbf{c}$, which is added as input to the network $\epsilon_\theta$. For us, $\mathbf{c}$ is the visual scene captured by one or more calibrated RGB-D images, a language instruction, as well as the proprioceptive information. Given a collection of $\mathcal{D} = \{(x^i, \mathbf{c}^i)\}_{i=1}^N$ of end-effector trajectories $x^i$ paired with observation and robot proprioceptive context $\mathbf{c}^i$, the denoising objective becomes:

$$\mathcal{L}_{\text{diff}}(\theta; \mathcal{D}) = \frac{1}{|\mathcal{D}|} \sum_{x^i, \mathbf{c}^i \in \mathcal{D}} ||\epsilon_\theta(\sqrt{\bar{\alpha}^i}x^i + \sqrt{1-\bar{\alpha}^i}\epsilon, \mathbf{c}^i, i) - \epsilon||. \quad (2)$$

This loss corresponds to a reweighted form of the variational lower bound for $\log p(x|\mathbf{c})$ [32].

In order to draw a sample from the learned distribution $p_\theta(x|\mathbf{c})$, we start by drawing a sample $x^N \sim \mathcal{N}(\mathbf{0},\mathbf{1})$. Then, we progressively denoise the sample by iterated application of $\epsilon_\theta$ $N$ times according to a specified sampling schedule [32, 64], which terminates with $x^0$ sampled from $p_\theta(x)$:

$$x^{i-1} = \frac{1}{\sqrt{\alpha^i}}\left(x^i - \frac{\beta^i}{\sqrt{1-\bar{\alpha}^i}}\epsilon_\theta(x^i, i, \mathbf{c})\right) + \frac{1-\bar{\alpha}^{i+1}}{1-\bar{\alpha}^i}\beta^i\mathbf{z}, \quad (3)$$

where $\mathbf{z} \sim \mathcal{N}(\mathbf{0},\mathbf{1})$.

### B.7 The importance of noise scheduler

Existing diffusion policies [21, 6] often deploy the same noise scheduler for estimate of position and rotation. However, we empirically found that selecting a good noise scheduler is critical. We visualize the clean/noised 6D rotation representations as two three-dimensional unit-length vectors

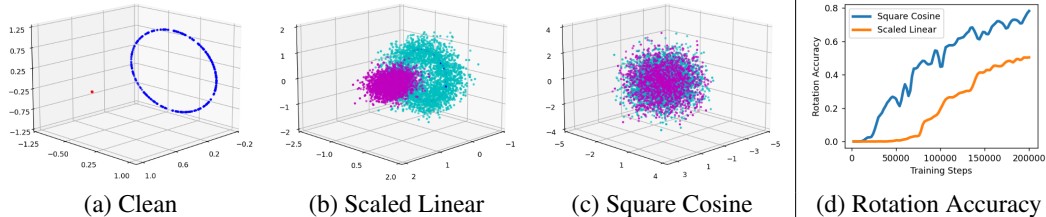

| (a) Clean | (b) Scaled Linear | (c) Square Cosine | (d) Rotation Accuracy |

Figure 7: **Visualization of noised rotation based on different schedulers.** We split the 6 DoF rotation representations into 2 three-dimension unit-length vectors, and plot the first/second vector as a point in 3D. The noised counterparts are colorized in magenta/cyan. We visualize the rotation of all keyposes in RLBench *insert_peg* task. From left to right, we visualize the (a) clean rotation, (b) noisy rotation with a scaled-linear scheduler, and (c) that with a square cosine scheduler. Lastly, we compare (d) the denoising performance curve of two noise schedulers. Here, accuracy is defined as the percentage of times the absolute rotation error is lower than a threshold of 0.025. Using the square cosine scheduler helps our model to denoise from the pure noise better.

in Figure 7. We plot each vector as a point in the 3D space. We show that noised rotation vectors generated by the squared linear scheduler cover the space more completely than those by the scaled linear scheduler, resulting in better performance than a square cosine scheduler

