# OpenReview forum: "3D Diffuser Actor: Policy Diffusion with 3D Scene Representations"
_robot-learning.org/CoRL/2024/Conference — CoRL 2024_

### Official Review · Reviewer_BGu2 · 2024-06-26
**Good, well-written paper**

**Originality:** 3
**Technical Quality:** 4
**Clarity Of Presentation:** 5
**Potential Impact:** 3
**Recommendation:** 3
**Confidence:** 5

**Review:**

Good, well-written paper with thorough evaluations. The paper is mainly an integration of two approaches: relative 3d transformer which has translationally invariant attention layers, and diffusion policy.

Strengths:
1. Strong results on the RLBench and CALVIN benchmarks demonstrating the importance of both 3d representations and diffusion models for performance.
2. Several ablations including comparison of policies trained from a single vs multiple RGBD views, as well as some analysis of policy robustness to noisy depth maps.
3. Thorough comparison to 6 baselines.
4. The paper is very clearly written with only minor details missing.
5. The method is likely to become an essential baseline for learning from demonstrations with RGBD observations.

Weaknesses:
1. The action representation feels unnatural for point clouds, see next section.
2. The main reason for a weak rather than a strong accept is that the work is an incremental integration of two existing approaches (relative 3d transformer and diffusion policies) and hence the contribution is not as original.

**Quality Of The Limitations Section:**

3

**Questions For Rebuttal:**

1. I find the choice of action representation a bit odd: singling out the discrete open/close action, and having separate noise schedules for the target orientation and position seems like a kludge. Have you considered representing actions as a bunch of 3d keypoints on the end-effector? That is a much more natural action representation for point clouds.

2. Why do you use horizon of length 1 for RLBench experiments? The ability to model trajectories is one of the main benefits of using diffusion, at the very least it would be good to show that your results hold when you predict longer trajectories.

3. Regarding comparison to DP3. Why did you choose to evaluate DP3 on CALVIN instead of RLBench? The DP3 paper doesn't consider language conditioning so it's unclear if this evaluation is meaningful. RLBench seems closer to the evaluations considered in DP3. (Note that this is not important for score given that DP3 was a concurrent work but it's good to clarify this.)

4. I couldn't find information about model sizes in the paper, please add those.

**Robotics Focus:**

4

**Summary Of Paper:**

The paper presents a novel architecture for diffusion policies operating on point clouds. They compare against a large selection of baselines, and show substantial improvements in success rates.

**Summary Of Recommendation:**

Accept

---

### Official Review · Reviewer_45HC · 2024-07-18

**Originality:** 3
**Technical Quality:** 3
**Clarity Of Presentation:** 4
**Potential Impact:** 3
**Recommendation:** 3
**Confidence:** 4

**Review:**

# Strengths

1. The integration of 3D scene representations with diffusion models is novel. The authors have successfully demonstrated how this combination can lead to improved policy learning for robotic manipulation tasks.

2. The authors have conducted extensive experiments, including comparisons with current SOTA policies and ablation studies. This thorough validation helps to build confidence in the proposed method's effectiveness. Besides, real robot experiments are also included.

3. The paper is well-structured, with a clear introduction, related work, methodological details, experiments, and conclusions. The figures and tables are informative.

# Weaknesses

1. The authors have shown impressive results on tasks with a parallel gripper. I am curious that whether 3D Diffuser Actor could also perform well on tasks with high-dimensional action space, such as dexterous manipulation tasks used in 3D Diffusion Policy? It would better show the generality of 3D Diffuser Actor.

2. Limited Discussion on Failure Cases: Although the paper briefly mentions failure cases, a more detailed analysis of why the model fails in certain situations and how these failures can be mitigated would be beneficial.

**Quality Of The Limitations Section:**

3

**Questions For Rebuttal:**

Please see weaknesses.

**Robotics Focus:**

4

**Summary Of Paper:**

The paper presents a novel approach to robot manipulation policies by combining 3D scene representations with diffusion objectives, named 3D Diffuser Actor. This approach has demonstrated significant performance gains over existing methods on benchmarks like RLBench and CALVIN, as well as in real-world applications.

**Summary Of Recommendation:**

I would possibly recommend accept since this paper presents strong empirical evaluations on 3D Diffuser Actor.

---

### Official Review · Reviewer_cssK · 2024-07-20
**Good contribution but some question/concerns**

**Originality:** 3
**Technical Quality:** 3
**Clarity Of Presentation:** 4
**Potential Impact:** 3
**Recommendation:** 4
**Confidence:** 4

**Review:**

Strengths:

The presented method works well, the paper is mostly well-written, and the experiments are well-executed. One can see that a lot of effort went into preparing the paper.

Weaknesses:

I do not see a major weakness in the contribution itself. Instead, I rather have some questions/concerns about some of the results and details contained in the appendix:
* The description of the 2D diffuser policy in lines 236/237 did not allow me to understand its implementation precisely.
* The benefit of the 3D diffuser architecture in single-view setups has not been tested. Table 3 is only concerned with the multi-view setup
* Why did the authors decide to employ 3D Diffusion Policies only in the CALVIN and not the RLBench benchmark? I'm asking because this seems to be the most closely related architecture.
* I could not understand the discussion around the temporal horizon in lines 282-286. Could the authors more clearly state the number of predicted actions per inference and the maximum number of executed actions? The current description mixes the concept of actions and trajectories.
* In line 290/291, the authors state that "allowing 3D Diffuser Actor to have a longer horizon before terminating increases the performance significantly, which suggests that the model learns to retry under failure." a) Can this longer horizon impact the fairness of comparison? b) How does the model perform with a shorter horizon? c) Did you verify that the model indeed retries under failure by inspection of rollouts?
* I disagree with the statement about control frequency. If the delay between observation and action is 600ms, the control frequency is 1.6Hz even if the prediction contains 6 endeffector poses. The simple reason is that the policy cannot react to unforeseen events during the 600ms delay. The number of predicted endeffecor poses is simply a parameterization choice. The policy could also predict 20 endeffector poses and would not be more reactive.
* The robustness evaluations in Appendix A.1 raise the question of whether training with noisy depth measurements could improve the robustness of the policy. Did the authors test such a training scheme?
* The authors should more clearly clarify that they are using a different model for the CALVIN benchmark (only stated clearly in Appendix 4b if I did not miss a statement in the main paper). Furthermore, I would like to see the performance of the "default" architecture in Figure 4a.

**Quality Of The Limitations Section:**

3

**Questions For Rebuttal:**

Improvements:
* A visualization of the 2D diffuser ablation similar to the visualizations in Figure 4 should be added.
* Please correct the statement about control frequency
* I would like to see ablations of the method in the single-view RLBench benchmark. After all, there may be many robotic tasks with a single view; hence, it is important to consider the architectural choices in this domain.
* I would like to see the performance of the "default" 3D diffuser policy presented in the main paper (in addition to the improved version).
* I would be happy if the authors can address my questions regarding the longer 3d diffuser horizon in the CALVIN benchmark.
* I would like to see an ablation against 3D Diffusion Policy in the stack cups task of the real-world evaluation as it seems to be one of the tasks with which the proposed methods struggles but which does not rely on language (please correct me if I am wrong here).

POST REBUTALL UPDATE:

The authors addressed all my points except for the last one. Hence I increased my score to a strong accept.

**Robotics Focus:**

4

**Summary Of Paper:**

The paper presents a novel architecture for diffusion policies that aims to better leverage 3D information often available in robotic applications while also allowing for language conditioning. Consequently, the authors opt for a transformer-based architecture that leverages spatial attention mechanisms. The method shows strong performance in experiments in the CALVIN and RLBench benchmarks.

**Summary Of Recommendation:**

As detailed, I like the paper except for some improperly presented findings and lacking ablations. If the authors can provide the missing details and fix the presentation accordingly, I am happy to increase my score to a strong accept.

---

### Author Rebuttal · Authors · 2024-08-12

We thank all reviewers for their constructive feedback. We  address all their concerns by replying to each reviewer individually.

Specifically:
1. We clarified all presentation questions and added one new figure (see the zip file).
2. We added further comparisons against DP3 which further validate the superior performance of our model.
3. We discussed how to extend our method to new robot morphologies, beyond parallel gripper.
4. We added an extended discussion of failure cases and suggested ways to mitigate them.
5. We argued that, while 3D Diffuser Actor builds upon existing work in 3D representation learning for robotics and diffusion policies, our technical contributions are significant, as indicated by our large quantitative gains over  related baselines that also combine 3D and diffusion.

We hope that our responses address all points raised by the reviewers and the AC. We are happy to provide further explanations and discussions if there are more concerns.

---

### Decision · Program_Chairs · 2024-09-04

**Decision:**

Accept

**Comment:**

Summarizing the weaknesses and strengths pointed out by the reviewers:

Strengths:
- The integration of 3D scene representations with diffusion models is novel
- Strong results on CALVIN and RLBenchmark
- Good ablation analysis
- Well-written
- Experiments are well-executed including extensive comparisons to baselines

Weaknesses:
- While well-written, some minor clarity issues (see reviews for specifics)
- Some architecture choices are not well-justified (e.g., why is 3D diffusion policies only used for CALVIN and not RLBenchmark?)
- Experiments are limited to a parallel gripper
- The discussion of failure cases is rather limited
- The methodology is an incremental integration of existing methods

Overall, all 3 reviewers recommend acceptance (though weakly) and have fairly positive things to say. This appears to be a fairly strong, well-executed paper. However, the authors should take care to address the reviewers's concerns as they point out many.

**Update after rebuttal**: The authors' rebuttal responded directly to the reviewers' concerns, including updating the paper. All 3 reviewers responded to the rebuttal, 2 choosing to maintain their recommendation of weak accept and 1 moving up to strong accept. Considering this, it seems prudent to recommend this paper be accepted.